# Molecular Pathways of WRKY Genes in Regulating Plant Salinity Tolerance

**DOI:** 10.3390/ijms231810947

**Published:** 2022-09-19

**Authors:** Lewis Price, Yong Han, Tefera Angessa, Chengdao Li

**Affiliations:** 1Western Crop Genetics Alliance, College of Science, Health, Engineering and Education, Murdoch University, Perth, WA 6150, Australia; 2Department of Primary Industries and Regional Development, Perth, WA 6151, Australia

**Keywords:** WRKY transcription factor, osmotic stress, ion detoxification, radical oxygen species, hormone signaling, salinity tolerance

## Abstract

Salinity is a natural and anthropogenic process that plants overcome using various responses. Salinity imposes a two-phase effect, simplified into the initial osmotic challenges and subsequent salinity-specific ion toxicities from continual exposure to sodium and chloride ions. Plant responses to salinity encompass a complex gene network involving osmotic balance, ion transport, antioxidant response, and hormone signaling pathways typically mediated by transcription factors. One particular transcription factor mega family, *WRKY*, is a principal regulator of salinity responses. Here, we categorize a collection of known salinity-responding *WRKYs* and summarize their molecular pathways. *WRKYs* collectively play a part in regulating osmotic balance, ion transport response, antioxidant response, and hormone signaling pathways in plants. Particular attention is given to the hormone signaling pathway to illuminate the relationship between *WRKYs* and abscisic acid signaling. Observed trends among *WRKYs* are highlighted, including group II *WRKYs* as major regulators of the salinity response. We recommend renaming existing *WRKYs* and adopting a naming system to a standardized format based on protein structure.

## 1. Introduction

Climate change is the most dangerous threat to humanity, with major implications for food production. One inherited issue from global warming is the increased salinization of arable land [1], which is a bottleneck for crop production. With the population set to reach nine billion by 2050 [2] and food production only meeting a fraction of what will be required, there is a deficit in predicted food availability [3]. As such, unproductive arid to semi-arid landscapes have been developed with irrigation systems to meet the food production demands. Irrigation schemes without adequate drainage, such as those in low rainfall areas, result in salinization, as evaporation brings salts throughout the profile to the surface via capillary action [4]. Developing salt-tolerant crops is required to match these conditions [5]. Understanding the tolerance mechanisms for improving existing varieties is key to overcoming the challenges presented [6].

Salinity stresses can be divided into two phases: (I) water gradient disruptions and (II) sodium ion (Na^+^) accumulation, which disrupt the plant’s physiological and biochemical functions. Interference in the water gradient has immediate consequences, including growth cessation, stomatal closure, sodium influx, and cell depolarization [7]. Continual exposure to Na^+^ results in their uptake; in toxic concentrations, Na^+^ disrupts cellular processes and causes an efflux of potassium ions from plant cells. For example, sodium binds competitively to potassium target sites, disrupting cell metabolism [8] and decreasing chlorophyll content and overall photosynthetic capabilities [9,10,11,12,13]. Therefore, salinity imposes complex challenges that require plants to employ a wealth of signaling pathways to overcome the barriers to normal function.

Upon detecting stress conditions, plants respond by directly enriching ion transporter activity or modulating regulatory pathways [14]. Transcription factors are regulators that target specific DNA regulatory elements directly, for positive or negative control [15], mediating the stress relief pathway. One particular transcription factor, WRKYs, regulate salinity through osmotic response, ion transport, oxidative stress relief, and hormone signaling pathways. *WRKY* genes are a common element in regulating biotic and abiotic stress responses [16,17], and a single WRKY protein will target specific W-box sequences which act as regulatory elements for downstream genes and subsequently induce many pathways.

In some species, over 100 *WRKYs* have been identified, some with roles in salinity tolerance. However, the positive and negative functions of *WRKYs* and their distinct salinity tolerance mechanisms have not been collated. In addition, the specific involvement of WRKY gene clusters in salinity tolerance remains unknown. This review presents a collection of known *WRKYs* that regulate the salinity response in plants and label them according to their pathways of action. In particular, *WRKY* genes involved in abscisic acid (ABA) signaling are scrutinized due to the high observed frequency for this pathway within *WRKY* genes. Observed trends between *WRKY* groups and the salinity response are outlined for future *WRKY* analysis when predicting salinity response. Further, we propose renaming existing *WRKY* genes based on an adopted naming scheme [18].

## 2. Mechanisms of Plant Tolerance to Salinity

Effective ion transport is a tolerance characteristic since excessive ion accumulation under salinity conditions, including Na^+^ and chloride (Cl^–^), can be toxic. Several key genes mediate this response. High-affinity potassium ion (K^+^) uptake transporter (HKT) mediates the continual uptake of K^+^ to ensure an optimal K^+^:Na^+^ ratio within the plant cells [19,20,21]. The salt overly sensitive (SOS) pathway involves Na^+^/H^+^ antiporters to maintain homeostasis, primarily involving *SOS1* but also *SOS2* and *SOS3* [16,22,23]. Another important intracellular Na^+^/H^+^ exchange is the *NHX*-type, with NHX proteins responsible for Na^+^ compartmentalization or sequestration into intracellular vacuoles [16,24].

The ability to adjust and maintain the osmotic gradient is also important under salinity stress. Organic solute production (e.g., proline) helps adjust the osmotic gradient and involves the pyrroline-5-carboxylate synthetase (P5CS) gene, among others [25]. ABA is a key hormone triggered under induced drought as it regulates stomatal opening [26] and proline accumulation. Lesser-known salt tolerance zinc finger (STZ) genes function as transcriptional repressors to improve the osmotic/drought response [27].

Reactive oxygen species (ROS) are a byproduct of cellular metabolism that drive ion transport and maintain an optimal osmotic gradient. Under salt stress, ROS production increases [28], affecting the delicate balance required for plant functioning by causing oxidative stress. Malondialdehyde (MDA) can accumulate in toxic levels under oxidative stress and is often used as a marker for lipid peroxidation under such conditions [29,30,31,32]. However, one study has disputed MDA accumulation as a marker for sensitivity, viewing it as a tolerance mechanism [33]. Managing ROS is important because one study reports a 40% reduction in thylakoid membrane proteins from oxidative effects [34], which subsequently reduces photosystem capability [34]. Combating ROS production requires antioxidant (AO) species that typically scavenge ROS. Common AO species include superoxide dismutase (SOD) [35], peroxidase (POD), catalase (CAT) [36,37], ascorbate peroxidase (APX) [38], glutathione S-transferase (GST) [39], and/or glutathione peroxidase (GPX) [37,38] which when accumulated will result in less oxidative harm to cellular processes.

Given that there are multiple overlapping mechanisms within a response pathway, signaling is delegated to ‘master switches’ such as hormones which can induce multiple mechanisms across multiple pathways in response to specific conditions. Numerous plant hormones, or phytohormones, play a critical role in salinity tolerance, including ABA, ethylene, salicylic acid (SA), jasmonic acid (JA), auxin, and gibberellic acid (GA) [40].

While all hormones play their part, ABA is the most important, crosstalking with other phytohormones and being responsible for various response mechanisms. ABA typically responds to osmotic stress but also plays a critical role in salinity tolerance and other abiotic stress signaling. For example, in angiosperms ABA regulates stomatal closure in response to osmotic stress [41], helps avoid pockets of salinity in the soil by retarding root growth from exploring such areas [21,42], and may be involved in ROS scavenging [40,43]. SA promotes growth under salt stress, activating proline accumulation, enhancing the antioxidant system, and improving photosynthesis [44]. SA and JA positively regulate salinity tolerance and crosstalk with other hormones: SA regulates GA, auxin, and ABA, while JA only mediates ABA [40]. However, excess SA reduces plant fitness compared to moderate levels that improve fitness and survivability [40]. JA primarily represses plant growth under salt stress, while promoting ion and ROS homeostasis. Other hormones involved in regulating the salinity response are ethylene, existing as a positive regulator, and auxin which is a negative regulator [40].

Hormones can be regulated by proteins or themselves regulate proteins that bind and specifically control gene expression (positive or negative) known as transcription factors, including NAC, MYB, bZIP, ERF/dehydration-responsive element binding protein (DREB), and WRKY [17]. Here, we outline *WRKY* genes involved in key regulatory mechanisms, including ABA regulation and the SOS pathway.

## 3. WRKY Genes

*WRKY* genes have proliferated predominantly in angiosperms but can be found in some slime, algae, and other organisms [45]. *WRKY* genes were first discovered in sweet potatoes [46]. The core structure of WRKY proteins is a 60 amino acid (aa) conserved WRKY sequence domain at the N-terminal and a Zn-finger motif at the C-terminal [47]. Generally, WRKY proteins bind to the DNA sequence motif (T)(T)TGAC(C/T), otherwise known as the W-box [47,48]. Selection between conserved W-boxes is based partially on neighboring sequences [49], but this is not universal as, for example, Hv-WRKY38 requires two W-box domains for effective binding [50]. Other WRKY proteins can diverge further with no evidence of W-box binding, such as NtWRKY12, which binds to a WK-box (TTTTCCAC) [51]. The C-terminal Zn-finger motif and N-terminal are responsible for the WRKY transcription factor’s ability to bind to DNA [52,53].

WRKY proteins are classified into four groups. Group I proteins comprise two domains, while Group II and III proteins comprise one domain [47,54]. Group I is further divided into subgroups Ia and Ib based on their zinc fingers (C_2_H_2_ zinc fingers and C_2_HC zinc fingers, respectively) [55]. Group II is subdivided into five groups (a–e) distinguished by amino acid motifs and phylogenetic analysis [47,54]. Group III classification is based on its divergent C_2_HC zinc finger, unique to the *WRKY* family [55]. Group IV WRKY proteins lack a zinc finger [56] and are divided into subgroups Iva, containing a partial zinc finger motif (CX_4_C), and IVb, containing no conserved Cys or His residues [56].

### 3.1. WRKY Genes in Biotic and Abiotic Response

*WRKY* genes are diverse in their function and consequently involved in the positive and negative regulation of various abiotic and biotic stresses. For instance, biotic regulators *HvWRKY1* and 2 play an important role in suppressing the pathogen-inducible gene, *HvGER4c*, which confers resistance to powdery mildew [57]. Silencing *WsWRKY1* in *Withania somnifera* decreased the activity of defense-related genes and reduced overall plant fitness upon fungal (*B. cinerea*) and bacterial (*P. syringae*) infection [58]. *OsWRKY62* negatively regulates basal defenses against pathogens, with its overexpression compromising *Xa21*, a basal defense gene, and thus reducing plant innate immunity [59]. Abiotic stresses are also under the purview of *WRKY* responses. In maize (*Zea mays*), *ZmWRKY106* was induced under drought and heat stress and weakly induced by salt stress [60]. OsWRKY30 helped alleviate the drought response in rice (*Oryza sativa*) by regulating relevant genes [61]. However, *WRKY* proteins are often associated with a complex web of interactions that enhance abiotic and biotic stress tolerance. In sugar cane (*Saccharum* spp.), *ScWRKY5* transcription was induced via inoculation with smut *Sporisorium scitamineum* (biotic stress) and by abiotic stresses such as polyethylene glycol (PEG) and NaCl [62]. In tomato (*Solanum lycopersicum*), *SlWRKY31* activated the plant defense mechanism upon multiple pathogen inoculations and was involved in drought and salt stress tolerance [63].

### 3.2. WRKY Genes Involved in Salinity Response

Currently, there are 94 *WRKY* genes identified in barley (*Hordeum vulgare*) [57], 83 in tomato [63], 103 in rice [64], and 62 in pepper (*Capsicum annuum*) [65]. Table 1 and Table 2 summarize known *WRKYs* that regulate salinity tolerance, including plant species, gene name, and tolerance mechanisms based on their expression, be it in the native host plant or ectopic expression in another species, such as Arabidopsis. However, *WRKY* genes are complex, with their response differing with salt concentration; Table 1 and Table 2 only report their signaling relationships in response to salinity.

This review collated and summarized *WRKYs* involved in regulating salinity tolerance from the literature. *WRKYs* were excluded if there was only evidence of being upregulated by salinity but no inferred tolerance or sensitivity mechanism. However, *WRKY* studies with no gene expression analysis but investigated protein accumulation were included selectively. *GmWRKY49* [115], *TaWRKY10-1* [116], Elaeis guineensis *WRKYs* [117], *MdWRKY100* [118], *IlWRKY2* [119], *WRKY75* [120], *GhWRKY25* [121] and a host of barley candidate genes [18,122] were excluded despite evidence of upregulation under saline conditions. Cherry rootstock (*Prunus avium* L.) *PaWRKY25*, *33*, and *38* were also excluded due to the lack of a pathway of inferred tolerance analysis despite some evidence of enhanced tolerance (e.g., enhanced chlorophyll accumulation and survival rates) [123]. *VvWRKY30*, *MxWRKY53*, *VuWRKY*, *GhWRKY46*, and *TaWRKY13* had a lack of gene expression analysis but evidence of protein accumulation to justify their inferred mechanisms. *TaWRKY2* [106] was not included due to inconsistencies within the literature. Two other papers [109,124] confused *TaWRKY2* with *TaWRKY1* (according to the former paper), containing an accession that was, confusingly, labeled *TaWRKY1*. In summary, while not appropriate for this review, the excluded papers are excellent starting points for future investigations. The next section explains the specific methods for categorizing *WRKYs* under different regulatory pathways.

### 3.3. Pathways for WRKY Mediating Salinity Response

Below is a summary of the frequency for inferring regulation for the investigated *WRKYs* (Figure 1). When investigating the effect pathway, a conserved mechanism cannot be assumed when inserted into different plants. Modeled from within *Arabidopsis*, Figure 2 illustrates the inferred mechanisms used to classify the different response pathways. Most *WRKY* papers include the transformation of *Arabidopsis* with some including tobacco, to examine the impact of transferring their selected *WRKY* gene. However, the returning data are not always translatable in the native plant; for example, *OsWRKY72* in rice improved salinity tolerance within the species [73] but reduced tolerance in *Arabidopsis* [72]. Similarly, *PcWRKY33* enhanced survival of its natural host (*P. cuspidatum*) under saline conditions but reduced overall fitness in transgenic *Arabidopsis* [75]. Hence the selection of *WRKY* to enhance tolerance is only possible through practical experimentation.

Distinguishing *WRKYs* that affect ion transporters is a fairly stringent and straightforward process (Figure 2a). Figure 3a summarizes *WRKYs* that infer regulation via the ion transport pathway. The *WRKYs* included here are those with evidence of differential gene regulation responsible for ion transporters. This includes the upregulation of the SOS pathway (*SOS1*, *SOS2*, and/or *SOS3*), *NHX*, and *HKT* genes for stabilizing and maintaining an optimal Na^+^/K^+^ ratio for cellular function. Discriminating factors included the irregular expression of transport genes from the presence of the relevant *WRKY* gene but not the observed optimal K^+^/Na^+^ ratios compared to wild type alone. An outlier was *SlWRKY3*, which enhanced various ion transport pathways, such as Na^+^/K^+^-transporting ATPase (NP000693), to improve salinity tolerance.

The oxidative stress relief pathway was discriminated based on the evidence of enhanced antioxidant (AO) upregulation and/or enriched AO protein accumulation. These genes included *POD*, *CAT*, *APX*, *GST*, *GPX*, and *SOD* pathways (Figure 2a), except for *MfWRKY70*, included in this review due to its in-depth analysis of the osmotic response pathway to overcome salinity, despite minor conclusion regarding *MfWRKY70’s* ability to overcome the ROS produced under salinity stress. *WRKYs* involved in the oxidative stress pathway are in Figure 3b.

The osmotic response pathway overlaps the ABA and hormone signaling pathways but warrants distinction. Figure 3c summarizes the pathways that determined what classifies osmotic response signaling. Distinguishing ABA signaling is discussed below and illustrated in Figure 2b. For this review, the criteria predominantly depended on the upregulation of *P5Cs* or *STZ*. Evidence of proline or soluble sugar accumulation alone was not used as a qualifying characteristic. Excluded *WRKYs* included *TaWRKY44* and *PbWRKY40* (accumulated organic sugar and proline), *VuWRKY*, *TaWRKY13*, *MxWRKY55*, *MxWRKY53, MbWRKY4* (accumulated proline), and *JcWRKY* (accumulated soluble sugars). The included *WRKYs* had evidence of specific gene enrichment in response to salinity stress for the relevant genes. *GmWRKY12* [125] was excluded because it lacked solute accumulation as evidence, but its promoter region included elements involved in numerous stress relief pathways, warranting its distinction.

Hormone signaling included *WRKYs* which were involved with ABA signaling and other hormones and is shown in Figure 3d. ABA signaling is strictly defined as outlined in the next paragraph and in Figure 2b, but classifying *WRKYs* for hormone signaling with other hormones was considered on a case-by-case basis. This loose definition for hormones other than ABA is outlined in Figure 2a.

Discriminating the *WRKYs* between the different forms of interaction within ABA signaling (Figure 3e) was tricky given the conflicting conclusions within the literature but is defined here and shown in Figure 2b. Signaling within the ABA pathway is divided into ABA-dependent or ABA-independent signaling. An established link exists between *WRKY* genes and the ABA pathway [126]; here, this relationship is specifically related to salinity. Conflicting conclusions exist for certain gene interactions, such as responsive to desiccation 29A (*RD29A*), recognized by different papers as dependent [127] and independent [81]. *MfWRKY70* was first reported to operate in the ABA-dependent pathway because *RD29A* was induced [96]; however, we concluded that *MfWRK70* is ABA-dependent due to the enrichment of the nine-cis-epoxycarotenoid dioxygenase (*NCED*) gene involved in ABA synthesis [128]. Using *RD29A* signaling to infer ABA-dependent signaling is not a good interpretation since it is induced by ABA-dependent and ABA-independent pathways [129,130,131,132]. Classifying *WRKYs* as ABA signaling was based upon evidence of interactions within the ABA signaling pathway, but not if the *WRKY* gene was induced by ABA signaling itself. In summary, ABA signaling was determined from interactions within the ABA signaling pathway and discriminating between dependent and independent pathways was based on the regulation of biosynthesis and selective literature definitions [131,132]. The classification of ABA signaling included the upregulation of gene families *NCED* [132,133], zeaxanthin epoxidase (*ZEP*) [131,134], aldehyde oxidase (*AAO*) [131,135,136], response to desiccation 29 (*RD29*), response to desiccation 22 (*RD22*) [130], *MYC* [137], *DREB*, abscisic acid-responsive elements-binding factor (*ABF*) [131], ABA insensitive (*ABI*) [131], and ABA-hypersensitive germination (*PP2CA*) [138,139]. Of these, only the *DREB* family was classified as ABA-independent signaling, while *NCED*, *ZEP*, *AAO*, *ABF*, *RD22*, and *ABI* were classified as ABA-dependent signaling due to roles in ABA biosynthesis, regulation, and what the selected literature distinguished as the ABA-dependent signaling pathway [131,132].

## 4. Standardizing WRKY Naming

The naming convention found for *WRKY* genes is not consistent within the literature and subsequently has limited practicality. Sometimes naming can be coincidental, as with *AhWRKY75* being most similar to *AtWRKY75* via homolog analysis [79]. Other times *WRKYs* are named due to their homology, as with *PcWRKY33* named after *AtWRKY33* [75] and *GmWRKY6* after *AtWRKY6* [71]. However, if there is no identified homolog with *Arabidopsis,* this system does not work, and the discovery sequence is then relied on for labeling. This mixture of systems has created inconsistent rules, undermining its practical use. *ZmWRKY17*, for instance, has close homology with *AtWRKY15*, *GmWRKY13*, and *VvWRKY11* but is not similar in numbered labeling [78]. Another example is *HvWRKY38*, with close homology with *AtWRKY40* and *OsWRKY71* [50]. Naming based on homology clashes with ascending numeration, such as *GhWRKY39-1* named according to homology with *AtWRKY39*; since *GhWRKY39* already existed, the authors’ solution was to denominate with ‘-1 [87]. Naming can also be distinguished by an isolated ruleset, such as *TaWRKY75-A* being labeled according to its location in the sub-genome [109]. This sub-genome labeling has been conserved due to its significance within the context of renaming here. In addition, musa*WRKY18* does not follow standard naming conventions [140] nor does *WRKY75* [120].

We propose renaming *WRKYs* based on their physical structure, adopting an existing renaming system [18] that was prompted while investigating *WRKYs* in the literature. Figure 4a, made from Table 1 and Table 2, show that most *WRKYs* are in group II, and as are the overwhelming majority of negative regulators as seen in Figure 4b. However, this does not match the literature, with abiotic stress believed to be linked proportionally more to groups I and III *WRKYs* [106].

Merging the trends observed from the *WRKY* grouping with a system built to incorporate the physical structure into the naming of a *WRKY* is a practical tool for future investigations. You cannot infer a mechanism by knowing its group, but it will be relevant when deciding the hierarchy for investigating *WRKYs*. An example of how the system by Yazdani, Sanjari [18] renames its *WRKYs* is *HvWRKY32*, a group III *WRKY* and, thus, subsequently named *HvWRKY_III11* with 11 denoting that this is the eleventh HvWRKY_III protein found. The *WRKYs* from Table 1 and Table 2 have been renamed with this adopted system, if possible, in Table 3. Some *WRKYs* were moved into other *WRKY* groups according to the phylogenetic tree in Figure 5. This tree was also used to classify ungrouped *WRKYs*, such as *AtWRKY8*, *GhWRKY17*, and *OsWRKY72*, if their accession was available; however, if no accession or group is given then they cannot be named under this practical system.

## 5. Conclusions

The challenges imposed by salinity require a network of responses to overcome, with the mechanisms varying between species and *WRKY* genes. With climate change predicted to further exacerbate dryland salinity [5], understanding the tolerance mechanisms employed by plants can help overcome these challenges. Salinity is summarized as a two-phase challenge of (I) imposed drought and (II) salinity-specific ion toxicities from extended exposure to NaCl ions. Numerous mechanisms are used to combat the challenges introduced by salinity, with *WRKY* transcription factors heavily involved [7]. Here, *WRKYs* have been labeled according to their signaling pathway in response to salinity, simplified into four categories: osmotic response, hormonal response, ion transport, and oxidative stress detoxification regulation. In the literature, salinity is managed primarily by group II *WRKYs*, and predominantly operate via the osmotic response pathway. Considerable crosstalk with ABA signaling (dependent or independent) also occurs, predominantly as a positive signaling pathway. Most *WRKYs* are positive regulators from group 2, and nearly all *WRKY* negative regulators were found to be from group II. The analyzed *WRKYs* were also included in a phylogenetic analysis, if possible, to determine their subgroup and confirm their grouping. We adopted a more practical naming system for *WRKY* genes to rename existing *WRKYs* in a coherent and standardized fashion that synergizes with the trends observed within this review.

This information will improve access to *WRKYs* involved in regulating salinity tolerance and accelerate investigations on hardier crop varieties. In this review, *WRKYs* involved in negative and positive regulation for salinity tolerance have been separated due to their difference in practicality. Identifying negative regulators is the most practical information for future crop varieties since there is no concern about foreign DNA insertions with gene knock out [142]. Negative *WRKYs* should not overshadow the practicality of positive *WRKYs,* however, as they can be transferred between species. The accessibility of such transformation is becoming easier because of ever-developing advanced technologies [6,143]. One such technology is the clustered, regularly interspaced, short palindromic repeat (CRISPR)/Cas systems which is making genetic investigations more accessible for fast-tracking potential new varieties [143,144].

## Figures and Tables

**Figure 1 ijms-23-10947-f001:**
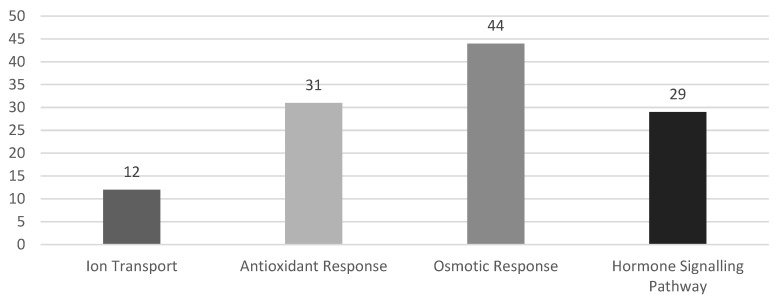
Bar graph visualizing the distribution of response pathways for *WRKY* genes.

**Figure 2 ijms-23-10947-f002:**
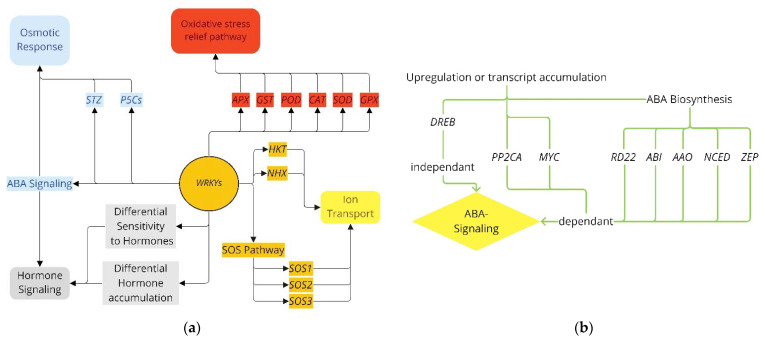
(**a**) Method for inferring the mechanism of action of differential regulation of genes *APX*, *GST*, *POD*, *CAT*, *SOD*, and/or *GPX* to determine involvement in the oxidative stress relief pathway. Differential regulation of *HKT*, *NHX*, and/or the SOS pathway to infer ion transport regulation. ABA signaling, differential hormone sensitivity, and/or differential accumulation due to the presence of *WRKY* are classified as hormone signaling. The osmotic response was according to differential expression of *P5Cs*, *STZ*, and ABA signaling. This is modeled after *Arabidopsis* since most *WRKYs* were investigated therein; (**b**) How *WRKYs* qualified for ABA signaling modeled in *Arabidopsis*. Split into independent signaling based on differential expression of *DREB* and dependent signaling based on differential regulation of ABA biosynthesis genes (*PP2CA* and *MYC*). Biosynthesis genes included *RD22*, *ABI*, *AAO*, *NCED*, and *ZEP*.

**Figure 3 ijms-23-10947-f003:**
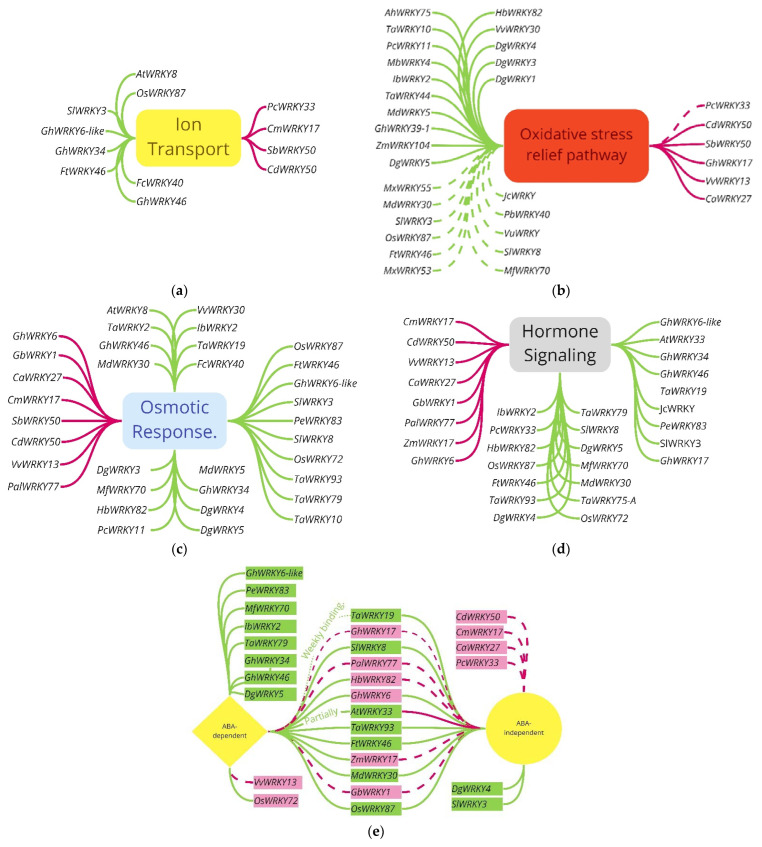
(**a**) *WRKYs* that positively (light green lines) and negatively (dark red) affect the ion transport response; (**b**) *WRKY* oxidative stress response pathway with positive (light green lines) and negative (dark red) interactions, solid lines indicate genotypic evidence, and dashed lines indicate phenotypic evidence; (**c**) *WRKYs* that regulate salinity tolerance through an osmotic response pathway with positive (light green lines) and negative (dark red) interactions. (**d**) *WRKYs* that regulate the hormone signaling pathway with positive (light green lines) and negative (dark red) interactions; (**e**) *WRKY* interactions with ABA signaling for dependent, independent, or both pathways; green boxes enhanced salinity tolerance, red boxes reduced salinity tolerance, dashed red lines indicate reduced expression, and solid green lines indicate enhanced expression of the ABA signaling pathway.

**Figure 4 ijms-23-10947-f004:**
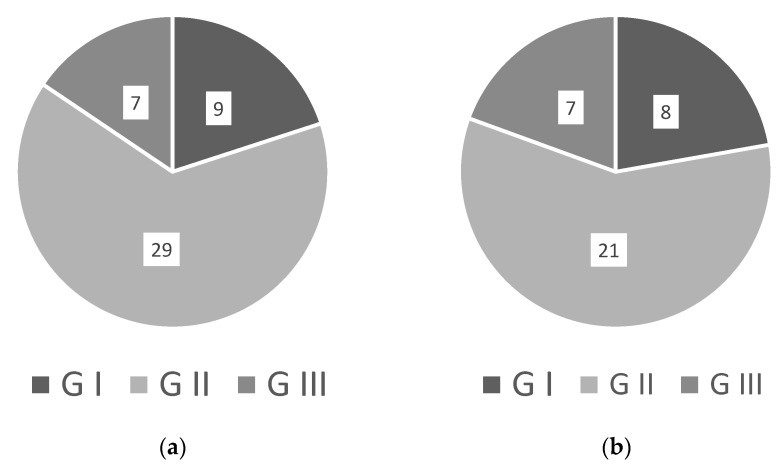
(**a**) Ratios of group I, II, and III *WRKYs* involved in the salinity response (see also Table 1 and Table 2); (**b**) Ratio of the groups containing *WRKYs* with a positive impact on salinity tolerance.

**Figure 5 ijms-23-10947-f005:**
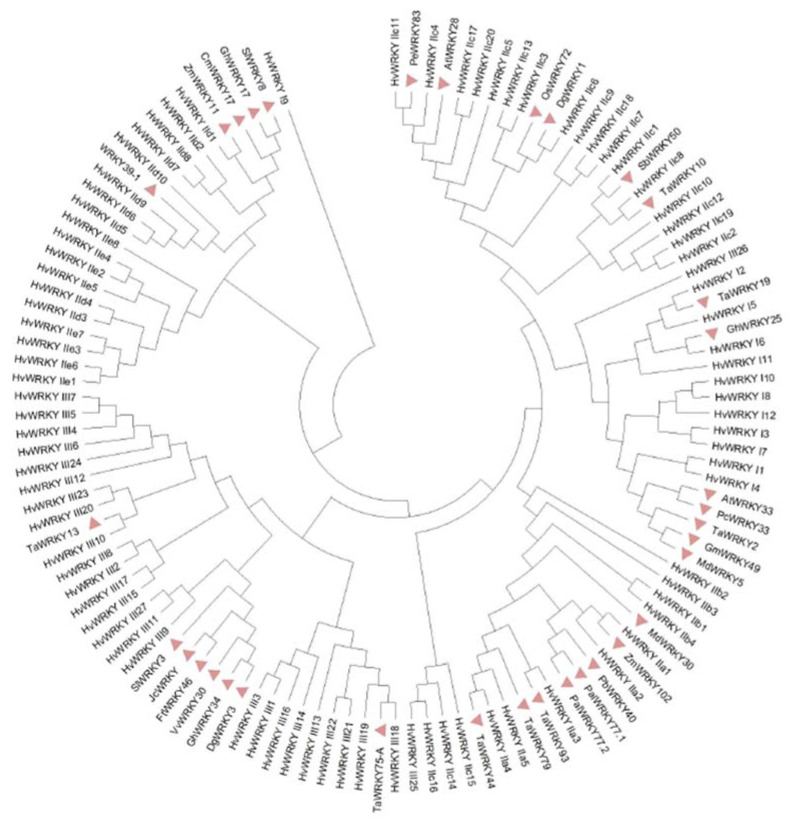
Phylogenetic tree used to confirm grouping and subgrouping of *WRKYs* from Table 1 and Table 2, outlined via indentation using MEGAX software [141].

**Table 1 ijms-23-10947-t001:** Negative WRKY transcription factor regulators for salinity response.

Plant (Species) Originate from	Expression Tested in	Gene ID	Protein ID	Function	References
**Pepper (*Capsicum annuum*)**	*Arabidopsis & Tobacco*	*CaWRKY27*	n.a.	Insertion reduced ROS-detoxification, hormone signalling, and osmotic response pathways	[66]
**Bermudagrass (*Cynodon dactylon* (L). Pers.)**	*Arabidopsis*	*CdWRKY50*	n.a.	Overexpression (OE) reduced hormone signalling, ion transport, ROS scavenging, and osmotic regulation pathways	[67]
**Chrysanthemum (*Chrysanthemum morifolium*)**	*Arabidopsis*	*CmWRKY17*	AJF11725 *	OE reduces hormone signalling and osmotic response pathways	[68]
**Cotton (*Gossypium barbadense*)**	*Arabidopsis*	*GbWRKY1*	n.a.	OE negatively regulated osmotic response and hormone signalling pathways	[69]
**Cotton (*Gossypium hirsutum*)**	Tobacco	*GhWRKY17*	ADW82098.1 *	OE enhanced sensitivity to saline conditions by reducing ROS regulation and hormone signalling pathways	[70]
**Cotton (*Gossypium hirsutum*)**	*Arabidopsis*	*GhWRKY6*	n.a.	OE reduced osmotic response and hormone signalling pathways	[71]
**Rice (*Oryza sativa*)**	*Arabidopsis* & rice	*OsWRKY72*	ALB35168.1 *	OE inhibited osmotic response and interfered with hormone signalling in *Arabidopsis*, but native expression enhanced rice salinity tolerance	[72,73]
**Poplar (*Populus alba* var. *pyramidalis*)**	*Populus alba var. pyramidalis*	*PalWRKY77*	Potri.003G182200.1 *, ⬢	Negative regulator reduced osmotic and hormone signal responses	[74]

Potri.003G182200.2 *, ⬢
**Japanese knotweed (*Polygonum cuspidatum*)**	*Arabidopsis* & native	*PcWRKY33*	AYN74370.1 *	OE reduced oxidative stress, osmotic response, and ion transport response pathways in Arabidopsis, but native expression enhanced salinity tolerance	[75]
**Sorghum (*Sorghum bicolor* (L.) Moench)**	*Arabidopsis*	SbWRKY50	Sb09g005700 **	OE reduced osmotic response, ROS scavenging, and ion transport pathways	[76]
**Grape (*Vitis vinifera*)**	*Arabidopsis*	*VvWRKY13*	n.a.	OE reduced ROS scavenging, osmotic response, and hormone signalling pathways	[77]
**Maize (*Zea mays*)**	*Arabidopsis*	*ZmWRKY17*	ACG39023.1 *	OE resulted in salt hypersensitivity and insensitivity to the ABA pathway	[78]

* NCBI GenBank database (https://www.ncbi.nlm.nih.gov/, accessed 14 April 2022), ⬢ unspecified between IDs, ** Plant Transcription Factor Database (TFDB, http://planttfdb.gao-lab.org/, accessed 16 April 2022).

**Table 2 ijms-23-10947-t002:** Positive observed WRKY transcription factors for salinity response.

Plant	Expressed in	Gene ID	Protein ID	Function	References
**Peanut** **(*Arachis hypogaea*)**	Peanut	*AhWRKY75*	n.a.	OE enhanced fitness and ROS scavenging	[79]
** *Arabidopsis thaliana* **	*Arabidopsis*	*AtWRKY33*	NP_181381.2 *	OE enhanced osmotic and hormone signaling pathways	[80]
** *Arabidopsis thaliana* **	*Arabidopsis*	*AtWRKY8*	NP_193551.1 *	OE enhanced osmotic response and ion transport pathways	[81]
**Chrysanthemum (*Dendranthema grandiflorum*)**	Tobacco	*DgWRKY1*	AGI96744.1 *	OE enhanced antioxidant response	[82]
**Chrysanthemum *(Dendranthema grandiflorum)***	Tobacco	*DgWRKY3*	AGN95658.1 *	Responsive to salt conditions, enhanced oxidative stress relief and osmotic response pathways	[83]
**Chrysanthemum (*Dendranthema grandiflorum*)**	Chrysanthemum	*DgWRKY4*	n.a.	OE enhanced ABA-independent pathways and ROS species	[84]
**Chrysanthemum (*Dendronthema grandiform*)**	Chrysanthemum	*DgWRKY5*	n.a.	OE involved in ABA signaling and pathway, ROS scavenging, osmotic regulator, and adjustment to infer salt stress tolerance	[29]
** *Fortunella crassifolia* **	Tobacco & Lemon	*FcWRKY40*	n.a.	OE enhanced osmotic response and ion transport pathways	[30]
**Tartary buckwheat (*Fagopyrum tataricum*)**	*Arabidopsis*	*FtWRKY46*	QGT76435.1 *	OE enhanced ROS scavenging and osmotic response and reduced hormone signaling	[85]
**Cotton (*Gossypium hirsutum*)**	*Arabidopsis*	*GhWRKY34*	AJT43314.1 *	OE enhanced hormone signaling, osmotic response, and ion transport pathways	[86]
**Cotton (*Gossypium hirsutum*)**	Tobacco	*GhWRKY39-1*	AGX27509.1 *	OE enhanced ROS detoxication pathway and enhanced fitness	[87]
**Cotton (*Gossypium hirsutum*)**	*Arabidopsis*	*GhWRKY46*	n.a.	Enhanced insensitivity to salinity through enhanced osmotic and ion transport response	[88]
**Cotton (*Gossypium hirsutum*)**	*Arabidopsis*	*GhWRKY6-like*	n.a.	OE enhanced ROS scavenging, osmotic response, and hormone signaling pathways	[89]
**Rubber tree (*Hevea brasiliensis*)**	*Arabidopsis*	*HbWRKY82*	n.a.	OE enhanced ROS scavenging, osmotic response, and hormone signaling pathways	[90]
**Sweet potato** **(*Ipomoea batatas* (L.) Lam.)**	*Arabidopsis*	*IbWRKY2*	n.a.	OE enhanced ROS scavenging, osmotic response, and hormone signaling pathways	[31]
** *Jatropha curcas* **	*Tobacco*	*JcWRKY*	AGE81984.1 *	OE enhanced ROS scavenging, osmotic response, and hormone signaling pathways	[91,92]
**Apple** **(*Malus baccata*)**	*Tobacco*	*MbWRKY4*	n.a.	OE enhanced antioxidant response and osmotic adjustment	[93]
**Siberian crab apple (*Malus baccata)***	Tobacco	MbWRKY5	MDP0000514115 **	OE enhanced membrane stability, osmotic response, and AO capabilities	[94]
**Apple (*Malus × domestica* borkh)**	*Arabidopsis &* Apple	*MdWRKY30*	QDL95022.1 *, ☐	OE enhanced ROS scavenging, hormone signaling, and osmotic response pathways	[95]
**Resurrection plant (*Myrothamnus flabellifolia*)**	*Arabidopsis*	*MfWRKY70*	n.a.	OE enhanced hormone signaling, ROS scavenging, and osmotic adjustment pathways	[96]
** *Malus xiaojinensis* **	*Arabidopsis*	*MxWRKY53*	n.a.	OE enhanced fitness, proline, and ROS scavenging activity	[97]
**Apple rootstock (*Malus xiaojinensis*)**	*Arabidopsis*	*MxWRKY55*	n.a.	OE enhanced ROS scavenging and osmotic response pathways	[98]
**Rice (*Oryza sativa*)**	*Rice*	*OsWRKY87*	n.a.	OE enhanced ion transport, osmotic response, and hormone signaling pathways and ROS-scavenging protein activity	[99]
**Southworth dance (*Pyrus betulaefolia*)**	*Arabidopsis*	*PbWRKY40*	Pbr004885.1 **	OE enhanced ROS scavenging and Na^+^ regulation via transporters	[100]
**Japanese knotweed (*Polygonum cuspidatum*)**	*Arabidopsis*	*PcWRKY11*	MZ734625 ****	OE reduced oxidizing elements and increased proline accumulation	[101]
**Moso bamboo (*Phyllostachys edulis; Bambusoideae*)**	*Arabidopsis*	*PeWRKY83*	PH01004514G0080 *	OE enhanced hormone signaling and osmotic response pathways	[32]
**Tomato (*Solanum lycopersicum*)**	*Arabidopsis*	*SlWRKY3*	ADZ15316 *	OE enhanced hormone signaling, osmotic response, ROS scavenging, and ion transport pathways	[102]
**Tomato (*Solanum lycopersicum*)**	*Solanum lycopersicum*	*SlWRKY8*	Solyc02g093050.2.1 *	OE enhanced osmotic response, ROS scavenging, and hormone signaling pathways	[103]
**Wheat (*Triticum aestivum*)**	Tobacco	*TaWRKY10*	ADY80578.1 *	OE enhanced osmotic response and ROS scavenging pathways	[104]
**Wheat (*Triticum aestivum*)**	Rice	*TaWRKY13*	Traes_2AS_ 6269D889E.1 **	Reduced ROS activity and enhanced proline accumulation in OE lines	[105]
**Wheat (*Triticum aestivum*)**	*Arabidopsis*	*TaWRKY19*	ACD80362.1 *	OE enhanced osmotic response pathway	[106]
**Wheat (*Triticum aestivum*)**	*Arabidopsis*	*TaWRKY2*	ACD80357.1 *	OE enhanced osmotic response pathway	[106,107]
**Wheat (*Triticum aestivum*)**	Tobacco	TaWRKY44	ALC04265.1 *	OE enhanced ROS tolerance and scavenging and compatible solute accumulation	[108]
**Wheat (*Triticum aestivum*)**	*Arabidopsis*	*TaWRKY75-A*	TraesCS4A01G193600.1 **	Involved in JA pathway	[109]
**Wheat (*Triticum aestivum*)**	*Arabidopsis*	*TaWRKY79*	AFN44008.1 *	OE enhanced hormone signaling and osmotic response pathways	[110]
**Wheat (*Triticum aestivum* L.)**	*Arabidopsis*	*TaWRKY93*	AFW98256.1 *	OE enhanced osmotic and hormone signaling pathways	[111]
**Bog bilberry (*Vaccinium uliginosum*)**	*Arabidopsis*	*VuWRKY*	n.a.	OE enhanced ROS scavenging and osmotic response pathways	[112]
**Grape (*Vitis vinifera* L.)**	*Arabidopsis*	*VvWRKY30*	ALM96663.1 *	OE enhanced osmotic response in proline accumulation and oxidative stress response activities	[113]
**Maize (*Zea mays*)**	Maize	ZmWRKY104	Zm00001d020495 ***	OE enhanced ROS scavenging response	[114]

* NCBI database (https://www.ncbi.nlm.nih.gov/, accessed 14 April 2022), ** within PlantTFDB (http://planttfdb.gao-lab.org/, accessed 16 April 2022), *** GrainGenes database (https://wheat.pw.usda.gov/GG3/, accessed on 17 April 2022), **** quoted to be in *GenBank* but doesn’t appear, ☐ labeled MdWRKY31 in NCBI GenBank.

**Table 3 ijms-23-10947-t003:** Proposed renaming of selected *WRKYs* from Table 1 and Table 2.

WRKY Gene	New Name	Issues with Naming and Comments
** *AhWRKY75* **	*AhWRKY_IIc1*	No protein ID available to check
** *AtWRKY33* **	*AtWRKY_I1*	–
** *AtWRKY8* **	*AtWRKY_IIc1*	No subgroup in the paper and thus deduced from phylogenetic analysis
** *CaWRKY27* **	–	No information on grouping and no accession available
** *CdWRKY50* **	–	Classed as group II but without subgroup and accession
** *CmWRKY17* **	*CmWRKY_IId*	No subgroup in the paper and thus deduced from phylogenetic analysis
** *DgWRKY1* **	*DgWRKY_Iic*	–
** *DgWRKY3* **	*DgWRKY_III1*	–
** *DgWRKY4* **	*DgWRKY_I1*	No protein ID available to check
** *DgWRKY5* **	*DgWRKY_I2*	No protein ID available to check
** *FcWRKY40* **	*FcWRKY_Iia*	No protein ID available to check
** *FtWRKY46* **	*FtWRKY_III1*	–
** *GbWRKY1* **	*GbWRKY_IIc1*	No protein ID available to check
** *GhWRKY17* **	*GhWRKY_IId1*	No subgroup in the paper and thus deduced from phylogenetic analysis
** *GhWRKY34* **	*GhWRKY_III1*	No protein ID available to check
** *GhWRKY39-1* **	*GhWRKY_IId2*	No subgroup in the paper and thus deduced from phylogenetic analysis
** *GhWRKY46* **	*GhWRKY_IIc1*	No protein ID available to check
** *GhWRKY6* **	–	No information on grouping and no accession available
** *GhWRKY6-like* **	–	No information on grouping and no accession available
** *HbWRKY82* **	*HbWRKY_IIc1*	No protein ID available to check
** *IbWRKY2* **	*IbWRKY_I1*	No protein ID available to check
** *JcWRKY* **	*JcWRKY_III1*	Classed as a group, but phylogenetic analysis deduced *JcWRKY* as group III
** *MbWRKY4* **	–	No information on grouping and no accession available
** *MdWRKY30* **	*MdWRKY_IIa1*	–
** *MbWRKY5* **	*MbWRKY_I1*	–
** *MfWRKY70* **	*MfWRKY_IIa1*	No protein ID available to check
** *MxWRKY53* **	*MxWRKY_IIc1*	No protein ID available to check
** *MxWRKY55* **	–	No information on grouping and no accession available
** *OsWRKY72* **	*OsWRKY_IIc1*	No subgroup in the paper and thus deduced from phylogenetic analysis
** *OsWRKY87* **	–	No information on grouping and no accession available
** *PalWRKY77* **	*PaWRKY_IIa1*	–
** *PbWRKY40* **	*PbWRKY_IIa1*	No subgroup in the paper and thus deduced from phylogenetic analysis
** *PcWRKY11* **	*PcWRKY_IId1*	No protein ID available to check
** *PcWRKY33* **	*PcWRKY_I1*	–
** *PeWRKY83* **	*PeWRKY_IIc1*	No subgroup in the paper and thus deduced from phylogenetic analysis
** *SbWRKY50* **	*SbWRKY_IIc1*	Grouped within class III but determined group II subgroup c from phylogenetic analysis
** *SlWRKY3* **	*SlWRKY_III1*	–
** *SlWRKY8* **	*SlWRKY_IId1*	–
** *TaWRKY10* **	*TaWRKY_IIc1*	Grouped within class I but determined to be group II subgroup c from phylogenetic analysis
** *TaWRKY13* **	*TaWRKY_III1*	Grouped within class II without subgrouping, but phylogenetic analysis determined to be group III
** *TaWRKY19* **	*TaWRKY_I1*	–
** *TaWRKY2* **	*TaWRKY_I2*	Grouped as group II but analysis determined closer to group I
** *TaWRKY44* **	*TaWRKY_IIa1*	Grouped as a class I protein, but analysis determined group II subgroup a
** *TaWRKY75-A* **	*TaWRKY_III2-A*	–
** *TaWRKY79* **	*TaWRKY_IIa2*	No subgroup in the paper and thus deduced from phylogenetic analysis
** *TaWRKY93* **	*TaWRKY_IIa3*	No subgroup in the paper and thus deduced from phylogenetic analysis
** *VuWRKY* **	*VuWRKY_I1*	No protein ID available to check
** *VvWRKY13* **	–	No information on grouping and no accession available
** *VvWRKY30* **	*VvWRKY_III1*	–
** *ZmWRKY104* **	*ZmWRKY_IIa1*	No subgroup in the paper and thus deduced from phylogenetic analysis
** *ZmWRKY17* **	*ZmWRKY_IId1*	–

## Data Availability

Data Sharing not applicable.

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
