# Peer review of "Molecular Pathways of WRKY Genes in Regulating Plant Salinity Tolerance"

_ijms, 2022, doi:10.3390/ijms231810947_

Round 1
Reviewer 1 Report
This review focused on WRKY genes involved in ABA signalling have been further dissected due to the high observed frequency for this pathway of effect within WRKY genes. There are also observed trends between the WRKY groups and salinity response outlined for future WRKY analysis when predicting salinity response. The review paper is well structured and well discussed. However, some points should be checked and corrected before its acceptance in this journal.
Therefore, according to my comments, I recommended the publication of the paper after major revision.
[1] Please provide the figure - Readers should easily understand the mechanisms.
[2] The study's background should be clearly stated. Describe the introduction and review of the work (Please add more information).
[3] In Conclusion, the authors should add the significance of this research and its potential practical application.
[4] The MS English needs to be improved. The article's English must be carefully checked for grammatical errors.
Author Response
Reviewer 1 Comments;
[1] Please provide the figure - Readers should easily understand the mechanisms.
We have added Figure 2 (a) and (b) which summarize what genes were used to classify the four different pathways, also a paragraph added to further illuminate the interpretation of the information;
Bellow of which a summary of the frequency for inferring regulation is seen in figure 1 (a) for the WRKYs investigated. However, inferring a universal mechanism is deceiving. When it comes to investigating the pathway of effect you cannot assume a conserved mechanism when inserted into different plants. Modeled from within Arabidopsis figure 2 (a) and (b) is the inferred mechanisms that were used to classify the different response pathways. Most WRKY papers involve the transformation of Arabidopsis, though also tobacco and others, to examine the impact of their focused WRKY gene if it’s transferal. But the returning data doesn’t always translate the predicted impact to its native plant as, for instance, the OsWRKY72 in rice improves salinity tolerance within the species [70] but in Arabidopsis reduced tolerance [69]. Another example is PcWRKY33 which enhances survival in salty conditions in P. cuspidatum, but when investigated in transgenic Arabidopsis resulted in reduced overall fitness [72]. Thus, although many genes presented here have evidence to say they enhance salinity tolerance, their transformations into different species cannot be predicted. Hence the selection of which WRKY for future candidates to enhance tolerance is only possible through practical experimentation.
|
(a) |
(b) |
Figure 2. (a) Method for inferring mechanism of action with differential regulation of genes APX, GST, POD, CAT, SOD, and/or GPX to classify as being involved in the oxidative stress relief pathway. Differential regulation of HKT, NHX, and/or the SOS pathway to infer ion transport regulation. ABA signaling, differential sensitivity to hormones, and or the differential accumulation due to the presence of the WRKY are classified as hormone signaling. The osmotic response was according to differential expression of P5Cs, STZ, and ABA signaling. This is modeled after Arabidopsis since the majority of WRKYs were investigated therein (b) How WRKYs qualified for ABA signaling modeled in Arabidopsis. Split into independent signaling based on differential expression of DREB and
[2] The study's background should be clearly stated. Describe the introduction and review of the work (Please add more information).
In the introduction a paragraph has been added to address the knowledge gap;
In some species there are over 100 WRKYs identified and the roles of some WRKY genes in salinity tolerance have been revealed. However, the positive or negative functions of WRKYs and their distinct salinity tolerance mechanisms has not been collected and summarized. Besides, the specific involvement of any WRKY gene clusters in salinity tolerance remains unknown. The purpose of this review is to collect and present the molecular pathways of WRKYs investigated and report their regulatory pathways.
[3] In Conclusion, the authors should add the significance of this research and its potential practical application.
Thank you! it is a good idea. The conclusion has been expanded upon with a single sentence (L395-397), and a second paragraph following the original (Greyed out text is the same, whilst black text is new);
The challenges imposed by salinity require a network of responses to overcome, and the mechanisms employed vary between species and subsequently WRKY genes. With climate change predicted to further enhance dryland salinity [5]- it’s by understanding the mechanisms of tolerance that these challenges can be overcome. Salinity itself can be summarized to impose a two-phase challenge of (I) imposed drought and (II) salinity-specific ion toxicities from extended exposure to NaCl ions. There are a plethora of mechanisms employed to combat the challenges introduced by salinity, and WRKY transcription factors are heavily involved in managing abiotic amongst other stresses [7]. WRKYs here have been labeled according to their signaling pathway in response to salinity, simplified into four categories such as osmotic response, hormone response, ion transport, and oxidative stress detoxification regulation. Analyzing the literature, it’s discovered that salinity is primarily controlled by group II WRKYs, and predominately will operate through the osmotic response pathway. There too is a lot of crosstalk with ABA, dependent or independent, signaling which dominantly is a positive signaling pathway. There are also observations made that the majority of WRKYs are positive regulators and that nearly all negative regulators are from group II WRKYs. Analyzed WRKYs were also involved in a phylogenetic analysis, if possible, to determine their subgroup and confirm their grouping. Adoption of an existing novel naming system for WRKY genes is used to rename existing WRKYs in a coherent and standardized fashion, that synergizes with the trends observed within this review.
The importance of this information is to make WRKYs involved in salinity more accessible and accelerate investigations towards hardier crop varieties. The isolation and presentation of WRKYS responsible for negative regulation of salinity tolerance were due to their importance. The identification of negative regulators is the most practical information for future crop breeding since there’s no concern over any insertion of foreign DNA. But that does not shadow the practicality of positive WRKYs, and they too hold importance for transferal within a species is possible and becoming more accessible with advancing technology [6, 141]. By summarizing WRKYs that regulate salinity with their accessions, more accessible phylogenetic studies can be conducted. The development of second-generation technologies such as clustered, regularly interspaced, short palindromic repeat (CRISPR)/Cas systems are making these kinds of genetic investigations more accessible, and the capability to fast-track potential new varieties and investigations into new ones is very real [141, 142]. In conclusion this review was made to provide insights into the genetic improvement of crop salinity tolerance through manipulation of WRKY genes from analyzing the literature.
[4] The MS English needs to be improved. The article's English must be carefully checked for grammatical errors.
Has been double-checked by a native speaker, and all the changes have been tracked
Reviewer 2 Report
Please find attached the file with my comments

Author Response
Reviewer 2 Comments;
The first paragraph must be improved. In my opinion, the authors should start by highlighting the current threat of using salt-contaminated water in irrigation as well as the problematic around soil salinization due to water evaporation. Then, I strongly advise the authors to avoid repeating words such as “disruption” and use more recent bibliography such as the review article about salinity “Parihar, P., Singh, S., Singh, R., Singh, V. P., & Prasad, S. M. (2015). Effect of salinity stress on plants and its tolerance strategies: a review. Environmental science and pollution research, 22(6), 4056-4075.”
Your suggestion for context highlights and oversight that has been corrected. In the introduction a new paragraph was added which include the impact of farming practices on soil salinization and the suggested article is cited in the text after;
Currently, climate change is the most dangerous threat posed to humanity, with major implications for food production. One particular inherited issue from global warming is an increase in salinification of arable land [1]. Salinity subsequently acts as a bottleneck to production which gets tighter over time. With the population set to reach 9 billion by 2050 [2], and food production only meeting a fraction of what’s required, there is a deficit in predicted food availability [3]. To meet the demands of food production, formally unproductive arid to semi-arid landscapes have been developed with irrigation systems. Irrigation schemes without adequate drainage, such as the aforementioned low rainfall areas, results in salinization as evaporation causes salts throughout the profile to be brought to the surface via capillary action [4]. If there is to be continual productivity in regions more adversely affected by climate change, the development of hardier crops is required to tolerate the aforementioned conditions [5]. Understanding the mechanisms of tolerance for the improvement of existing varieties is key to overcoming the challenges presented [6].
- 44-45: I think this sentence “…signaling response pathways employed to overcome the barriers presented by salinity” must end at pathways because this sentence starts with “salinity stress relief pathways”
(L:55-57) This error has been re-written to make sense, thank you for pointing this out;
The pathways through which WRKYs infer regulation of salinity can be simplified into osmotic response, ion transport, oxidative stress relief, and hormone signaling pathways
L.197-199/234-237: I don’t understand the meaning of these sentences. Please change it.
L.197-199 Was hard to read in reflection, thank you. This sentence has been removed.
L.234-237: This sentence for hormone signaling, along with the rest of the paragraph, has now been reworded into;
Hormone signaling has been loosely labeled based on a case-by-case basis per literature report, and this loose definition is reflected in figure 2 (a). If there was evidence for a change in hormone activity and/or specific genes enhanced which are involved in hormone regulation, such as being part of the biosynthesis pathway, then the WRKY qualified and are shown in figure 3 (d). The majority of WRKYs which were involved in hormone signaling involved ABA with very few exceptions, and thus how ABA signaling was discriminated won’t be repeated here. Discrepancies do exist since hormone sensitivity was included as a qualifying characteristic for hormones other than ABA, due to ABA being investigated in greater depth.
After reading the manuscript, I had serious doubts regarding its organization. For instance, why did the authors named “WRKY Genes Regulating Salinity Tolerance” if this section is mainly focused on the issues regarding the difficulty of understanding the mode of action of WRKY genes using distinct plants. I suggest that this section should be joint to the previous and another one, reporting those transformation and standardization name troubles, should be created.
The section title ‘WRKY Genes Regulating Salinity Tolerance’ has been deleted, but the content has been shuffled. The content outlining the issues with understanding the information has been moved to the beginning of ‘Pathways for WRKY mediating salinity response’ (L.231-245). The information regarding the trends for WRKY gene clusters has been moved to, ‘Standardizing WRKY Naming’ as reinforcing information (L.362-368). The paragraphs have the same information, but their wording has been altered to fit within context.
I would also like to suggest the authors to improve the Figures in order to make them more appealing.
Higher resolution images have been used and the figures changed to be more appealing. Every figure and their legends have been changed accordingly to reflect the line work and the color;
|
(a) |
(b) |
|
(a) |
(b) |
|
(c) |
(d) |
|
(e) |
|
Finally, the bibliography must be improved. Some references have a DOI while others don’t, and some species names are not in italic.
I reformatted the references into the annotated style, and the references combed for inconsistencies and have been corrected.
Other Changes.
Conducted minor changes to the text for the whole document, which have been tracked with ‘track changes.’ Otherwise, there has been the addition of more recent publications. Some references have been updated with latest publications
3.1 WRKY genes in Biotic and abiotic response.
L.184-185 extended a sentence (greyed text is unmodified, and the black text is new;
In tomatoes, SlWRKY31 has been observed to be an activator of plant defense mechanism upon multiple pathogen inoculations, but also evidence of being responsible for drought and salt stress tolerance
3.2. WRKY genes involved in salinity response.
Information has been shuffled here from elsewhere to above table 2, but the information is the same.
Table 2 has had GmWRKY12 removed and the addition of DgWRKY3.
After table 2 there is another paragraph which was moved from ‘Pathways for WRKY mediating salinity response’ and reworded and expanded (L210-229).
3.3 Pathways for WRKY mediating salinity response.
The paragraphs within this section have been updated to give more background information and be clearer. Otherwise, they have similar information with stricter definitions. The figures too have been updated to be more visually appealing.
- Standardizing WRKY Naming
There has been a decision change for TaWRKY75-A to be renamed to TaWRKY_III2-A so the sub genome designation is retained. The phylogenetic tree was also updated to be more visually appealing.
Round 2
Reviewer 1 Report
Requested corrections were completed.
Author Response
Thanks for your critical comments and suggestions. We have carefully revised the MS with proofreading from a professional editor. The track change document is attached

Reviewer 2 Report
I think that this manuscript has been substantially improved. However, I still detected some English errors that should be corrected before publication
Author Response
Thanks for your critical review and comments. We have thoroughly edited the MS with final proofreading by a professional editor. A track-changed document is loaded for your reference